**Data Availability Statement:** Data access is restricted as the data relevant to this study contains potentially identifying and sensitive patient

# Point of Care Lung Ultrasound Injury Score—A simple and reliable assessment tool in COVID-19 patients (PLIS I): A retrospective study

**Lior Fuchs**[1,2,3⊙]*, **Ori Galante**[1,3⊙], **Yaniv Almog**[1,3], **Roy R. Dayan**[3,4], **Alexander Smoliakov**[3], **Yuval Ullman**[3], **David Shamia**[3,4], **Ran Ben David Ohayon**[3,4], **Evgeny Golbets**[3,4], **Khaled El Haj**[3,4], **Jonathan Taylor**[5], **Itai Weissberg**[3,4], **Victor Novack**[2,3], **Leonid Barski**[3,4], **Eli Rosenberg**[2,3], **Eyal Gohar**[3], **Muhammad Abo Abed**[3,4], **Iftach Sagy**[2,3]

**1** Intensive Care Unit, Soroka University Medical Center, Beer-Sheva, Israel, **2** Clinical Research Center, Soroka University Medical Center, Beer-Sheva, Israel, **3** Faculty of Health Sciences, Ben-Gurion University of the Negev, Beer-Sheva, Israel, **4** Internal Medicine Division, Soroka University Medical Center, Beer-Sheva, Israel, **5** Department of Medicine, Oregon Health & Science University, Portland, Oregon, United States of America

⊙ These authors contributed equally to this work.

* liorfuchs@gmail.com

## Abstract

### Background

In COVID-19 patients, lung ultrasound is superior to chest radiograph and has good agreement with computerized tomography to diagnose lung pathologies. Most lung ultrasound protocols published to date are complex and time-consuming. We describe a new illustrative Point-of-care ultrasound Lung Injury Score (PLIS) to help guide the care of patients with COVID-19 and assess if the PLIS would be able to predict COVID-19 patients' clinical course.

### Methods

This retrospective study describing the novel PLIS was conducted in a large tertiary-level hospital. COVID-19 patients were included if they required any form of respiratory support and had at least one PLIS study during hospitalization. Data collected included PLIS on admission, demographics, Sequential Organ Failure Assessment (SOFA) scores, and patient outcomes. The primary outcome was the need for intensive care unit (ICU) admission.

### Results

A total of 109 patients and 293 PLIS studies were included in our analysis. The mean age was 60.9, and overall mortality was 18.3%. Median PLIS score was 5.0 (3.0–6.0) vs. 2.0 (1.0–3.0) in ICU and non-ICU patients respectively (p<0.001). Total PLIS scores were directly associated with SOFA scores (inter-class correlation 0.63, p<0.001), and multivariate analysis showed that every increase in one PLIS point was associated with a higher risk

information. The data are owned by the Clalit Health Services and these restrictions are imposed by the local ethics committee. For further details and to request access to the data please contact NaomiAm@clalit.org.il, the head of the research unit at Soroka University Medical Centre.

**Funding:** The authors received no specific funding for this work.

**Competing interests:** LF is a consultant of General Electric healthcare. This does not alter our adherence to PLOS ONE policies on sharing data and materials.

**Abbreviations:** AKI, Acute kidney injury; ARDS, Adult respiratory life support; COVID19, coronavirus disease 2019 (COVID-19); CT, Computed Tomography; Fio2, Fractional inspired oxygen; ICU, Intensive Care Unit; LUS, Lung Ultrasound; PaO2, Partial pressure of oxygen in arterial blood; PLIS, Point of Care Lung Ultrasound Injury Score; SOFA, Sequential Organ Failure Assessment.

for ICU admission (O.R 2.09, 95% C.I 1.59–2.75) and in-hospital mortality (O.R 1.54, 95% C.I 1.10–2.16).

## Conclusions

The PLIS for COVID-19 patients is simple and associated with SOFA score, ICU admission, and in-hospital mortality. Further studies are needed to demonstrate whether the PLIS can improve outcomes and become an integral part of the management of COVID-19 patients.

## Background

The novel coronavirus disease 2019 (COVID-19) pandemic is a global crisis, challenging healthcare systems worldwide. COVID-19 patients may present with profound hypoxemia without accompanying respiratory distress, also called "happy hypoxemia" [1]. The ambiguous clinical presentation, combined with the unpredictable course and potential for rapid deterioration, mandates an objective, quick, bedside lung assessment tool that helps assess illness severity and guide clinical decisions [2].

There is a direct association between the severity of COVID-19 pneumonia and computerized tomography (CT) findings. Ground glass opacities, lung consolidations, and peribronchial thickening are common [3]. CT scans may be unsafe and challenging to perform in severely hypoxemic patients that require meticulous isolation measures. In COVID-19 patients, lung ultrasound (LUS) was found superior to chest X-ray in detecting lung pathologies [4, 5] and has good agreement with chest CT, suggesting LUS may serve as a safe and convenient alternative that can be performed bedside [6, 7].

The characteristic LUS findings in COVID-19 involve B-lines, consolidations, and the pleura. B-lines can appear as focal, multifocal, or confluent, and consolidations present in various patterns, including multifocal, non-trans lobar, and trans-lobar with air bronchograms [3, 8, 9]. The pleural line appears thickened and irregular, and pleural effusions are uncommon. These findings collectively are associated with mortality and Intensive Care Unit (ICU) admission [10, 11]. Moreover, among COVID-19 patients, consolidations were specifically related to critical illness [12].

Most LUS protocols published to date are cumbersome and time-consuming. They typically involve screening at least 12 different lung areas, each graded from 0 to 3 points, thus generating scores ranging from zero to 36 [6, 11–14]. While these scores are informative for research purposes, they are less practical during a pandemic when physicians need to assess many severely ill patients under restrictive personal protective equipment and time constraints. In addition, the simplistic single numerical score fails to illustrate the location of lung findings, denying spatial information from the clinical team. Moreover, in these previously reported systems, B-lines are graded and given the same final weight in nondependent lung fields as dependent areas, where the specificity of these findings is limited, while consolidations are only reported in a yes/no binary format without size or localizing information. These research-focused protocols restrict many of the LUS advantages and make LUS use for daily clinical care impractical. Therefore, our investigative group developed a fast and straightforward LUS score.

Here we present a novel Point-of-care ultrasound Lung Injury Score–called the PLIS—for COVID-19 patient assessment. PLIS involves only three scanning areas on each lung, based on the known BLUE protocol scanning points, and grade the lung pathologies on a 1–6 scale.

This report is the first description and implementation of the PLIS. We hypothesized that using this focused clinical-sonographic score in admitted COVID-19 patients' assessment would be associated with the need for ICU admission.

## Methods

This is a retrospective study, data were collected during patients assesments and rounds. The PLIS protocol was taught and implemented in the COVID 19 wards and ICUs, and was part of the daily pateints' assessment. PLIS was assessed multiple times among patients admitted with COVID-19 between April 1st and June 30th, 2020, to Soroka University Medical Center in Israel. The study period included the Alpha, and Beta SARS-CoV-2 variants.

The study was approved by the hospital's ethics committee (IRB #0195–20). The PLIS protocol was developed by the authors at the beginning of the outbreak. Scans were carried out during most morning rounds and during clinical deterioration, as detailed below. The results of each PLIS study were documented in the electronic medical record. Patients were included in the analysis if they were admitted to the hospital, had a confirmed diagnosis of COVID-19, required any method of respiratory support (i.e., nasal cannula, non-invasive ventilation, invasive ventilation, etc.), and had at least one PLIS study. The ultrasound machines used were VENUE GO, GE Healthcare, R2 version. PLIS was conducted with the 3SC probe, using the manufacturer's lung preset to detect B-lines and the cardiac preset to detect consolidations. The PLIS was assessed bedside by internal medicine residents and senior physicians. The residents, who already had a basic knowledge of performing point-of-care LUS, received two hours of hands-on training specifically on the PLIS protocol and its components. The senior physicians were all intensivists with over seven years of experience in LUS; all were the designers of this score. While in the non-ICU settings, PLIS was performed solely by internal medicine residents, the exam was conducted by both residents and senior physicians evenly within the ICU. Inter-observer reliability, calculated by comparing lung scan reads from 2 different operators on 16 different pateints, was assessed in a blinded manner and was found adequate (Cohen's kappa 0.607, 0.750).

Data collected included the PLIS recorded on admission, when conducted during ward/ICU rounds, and when otherwise clinically indicated. As this is not a prospective study, the PLIS exams were not protocolized. Some patients could receive a PLIS study on regular ward admission, during the morning rounds, and also later, when admitted to the ICU. Some received ultrasound lung scan only during the ICU admission or only in the regular ward. Criteria for ICU admission included (1) patients who suffered respiratory failure and required mechanical ventilation or (2) those who needed non-invasive ventilation (high flow nasal cannula or bilevel positive pressure ventilation) and were assessed by an intensivist to potentially require mechanical ventilation in the next 24 hours potentially. Other variables included demographic characteristics, relevant laboratory tests, medical history, vital signs, and ventilation parameters. Outcomes measured included: daily SOFA score, length of admission, ICU admission, mortality, days of mechanical ventilation, and the ratio between arterial oxygen partial pressure and fractional inspired oxygen ($PaO_2/FiO_2$ ratio). The primary outcome was the requirement for ICU admission. Secondary outcomes were in-hospital mortality and a composite outcome composed of in-hospital mortality, prone position, and prolonged mechanical ventilation (>14 days). This study aimed to assess whether the PLIS is associated with the severity of illness, whether it is feasible to perform at the bedside in a short time, and if it may predict the clinical course of patients with COVID-19.

## Lung ultrasound—The PLIS

Lung ultrasound was conducted on patients in the supine, semi-supine, or prone position in three areas; over the midclavicular line on the upper and lower thorax (zone 1, upper and lower regions) and above the diaphragm in the mid and posterior axillary line (zone 2, lower area) (Fig 1a). The total PLIS comprises three separate elements, labeled as the A, B, and C

**a**

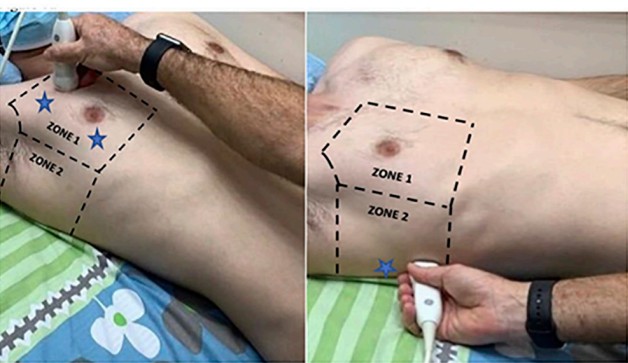

**b : Zone 1**

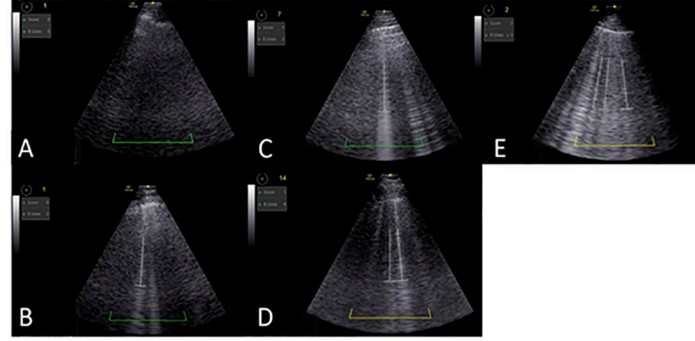

A and B: PLIS B 0 (less than 3 b lines )
C and D: PLIS B 1 (3-5 b lines )
E: PLIS B 2 (over 5 b lines)

**c : Zone 2**

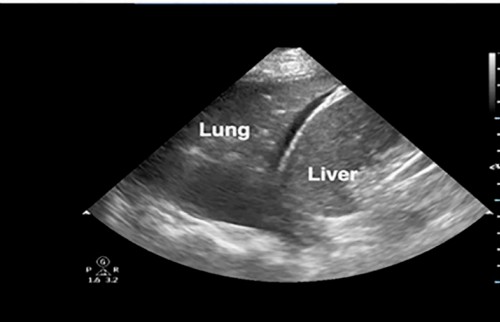

PLIS C 2 (consolidation over 4 cm)

**Fig 1. a.** Lung zones: The three scanning areas are marked with blue stars. **b.** Zone 1, PLIS B, **c.** Zone 2, PLIS C. The stars represent the area of the scan in each zone and not the exact place of the probe.

**Table 1. The Point of Care Lung Ultrasound Injury Score (PLIS) grading system.**

| SCORE | 0 | 1 | 2 |
|---|---|---|---|
| A | Room air/nasal prongs | Any non-invasive support over nasal prongs | Intubated or extracorporeal membrane oxygenation |
| B* Zone 1 only -upper and lower areas | B-lines <3 | B-lines 3–5 | B-lines > 5 |
| C Zone 1—upper and lower areas Zone 2- lower | No consolidation | Small consolidation (either unilateral or bilateral) | Large** consolidation (either unilateral or Bilateral) |

* B-Lines contribute to the score only if located bilaterally (The higher number of B-lines from any side defined the grading of the B component)

**Large consolidation: over 4 cm measured from the largest diameter.

components of the PLIS (Table 1). Each is graded from 0 to 2. The PLIS was designed to combine the method of respiratory support (the A score) with a report of the two major COVID-19 ultrasonographic findings: the interstitial alveolar syndrome, diagnosed by bilateral B-lines (the B score), and lung consolidations (the C score). The final PLIS score comprises the sum of the three components, ranging from 0–6 points.

**A score—The respiratory support.** The "A" component reflects the method of respiratory support that the patient is receiving. It incorporates into the PLIS an objective, quantifiable measure of the severity of respiratory failure based on the modality needed to maintain appropriate oxygen levels. PLIS scores room air/nasal cannula as A0. More intensive support (e.g., non-rebreather mask, high-flow nasal cannula, or non-invasive positive-pressure ventilation) grades A1. The need for invasive mechanical ventilation, prone positioning, or the requirement for extracorporeal membrane oxygenation meets the criteria for A2.

**B score- interstitial syndrome.** Multifocal bilateral B-lines, either discrete or confluent, are predominant in COVID-19 [6, 9]. Up to 3–5 B-lines, known as septal rockets [15–17], are graded as B1. The presence of either ground-glass rockets (i.e., > 5 B-Lines) or confluent and fused B-lines are graded as B2. In the PLIS, B-lines are counted only in the anterior thorax (midclavicular line, upper and lower zone 1, Fig 1b) as posterior interstitial changes can be found incidentally due to gravity alone. Additionally, B-lines only contribute to the score if located bilaterally to avoid consideration of artifacts [18]. The higher number of B-lines from any side defined the B component score of the PLIS.

**C score- lung consolidations.** As COVID-19 progresses, patients develop bilateral, peripheral-predominant ground-glass opacities, consolidations, or both [3, 9]. The majority of lung consolidations touch the pleura and are visible on LUS, with most cases localizing to the posterior-inferior lung areas [19]. The PLIS evaluates for consolidations over zone 1 and zone 2 (Fig 1a and 1c). A C-score between 0 to 2 is given by the extent of consolidations (Table 1) with a size cut-off set at 4 cm in the largest dimension. A score of 0 is assigned if no consolidation is appreciated. A score of 1 indicates smaller consolidations (< 4 cm), while a C-score of 2 describes a significant consolidation (≥ 4 cm). For localization purposes, unilateral consolidations, whether large or small, specifically received the letter "R" (for right-sided) or "L" (for left-sided) attached to the numerical score. Bilateral consolidations with at least one classifying as large are labeled as C2.

**Putting it together.** The total PLIS includes the sum of A, B, and C components. For example, the calculated PLIS for a patient on non-invasive ventilation, with more than five B-lines (located in zone 1 and bilaterally, Fig 1b-panel E) and small bilateral consolidations (situated in zone 1 or zone 2), is A1B2C1 with a total PLIS of 4. A patient receiving oxygen supplied by nasal cannula, with over five B-lines located only over the right lung and a right large

consolidation situated in zone 2 (Fig 1c), will have a PLIS of A0B0C2R, resulting in a total PLIS of 2. In this second example, the B-lines did not confer any points to the PLIS B score, as they were not detected bilaterally and thus more likely reflected an artifact from consolidation and not from the COVID-19 lung interstitial syndrome. The PLIS is designed to provide clinicians with information on the severity of the respiratory failure (A score) along with an image of the pathologic findings (B and C scores), thereby integrating clinical, physiologic data into a visuospatial representation of the LUS extent of affected lung. Ultimately, the total PLIS results in an overall assessment of COVID-19 severity and risk for worsening disease course.

**Statistical analysis.** Data were analyzed using SPSS 25.0. Data were expressed as mean ± standard deviation (SD), median ± interquartile range (IQR), or number and percentage. The unit of analysis was a single LUS test per patient. Patient characteristics were compared between patients admitted to an ICU versus non-ICU patients using the t-test, chi-square, and non-parametric tests. Inter-observer reliability was assessed by comparing independent ratings of the PLIS B and C scores among 16 randomly selected patients in a blinded manner. The comparison was made between a senior physician with seven years of lung ultrasound experience (LF) and four rating residents. The PLIS was compared to the SOFA score to assess external validation of the former. SOFA score was calculated for the same day per each LUS test. Correlation between each patient's PLIS and SOFA scores was calculated using Spearman's rank correlation coefficient and inter-class correlation. To assess internal validation, Cohen's Kappa was calculated for 16 random PLIS, evaluated by two experienced independent physicians. Multivariate generalized estimating equation (GEE) regression evaluated covariates associated with primary and secondary outcomes. The final model was selected based on the plausible clinical explanation, statistical significance, and goodness of fit using c-statistics.

## Results

A total of 109 patients and 293 PLIS studies were included in the analysis. Table 2 depicts the patient's baseline characteristics and outcomes. The mean age was 60.9 years (±13.6). About two-thirds of cases (n = 76, 69%) were males, 32% (n = 34) suffered from diabetes mellitus, and the mean body mass index was 28 (±5.4). The median hospitalization length was nine days in total, and the overall mortality was 18.3%. The mean time of PLIS scan evaluation was 3:54 (±1:07) minutes. The Cohen's kappa for inter-observer reliability was 0.607 for the B component and 0.750 for the C component.

A statistically significant direct association between each of the PLIS components (A, B, and C) and the SOFA score is shown in Fig 2A (PLIS A p<0.001, PLIS B p = 0.001, PLIS C p = 0.001). Fig 2B demonstrates a similar association between total PLIS (summation of A, B, and C) and the SOFA score (interclass correlation 0.63, p<0.001). Fig 3 further emphasizes the relation between PLIS and clinical outcomes by depicting an association between both higher initial and worst PLIS during hospitalization with ICU admission (A) and in-hospital mortality (B) (P<0.001 for all pairs).

Table 3 depicts the differences in clinical characteristics between ICU and non-ICU patients. The mean number of PLIS studies conducted for ICU patients was 5.3 versus 1.4 for non-ICU patients. The SOFA score was predictably higher in the ICU group (median 0 versus 5, p<0.001). For all PLIS components, findings of A, B, and C, grade 0 (i.e., lower score) were significantly more prevalent among the non-ICU ward patients, while grades 1 and 2 were much more prevalent among scanned performed on ICU patients. The median total PLIS was 2.0 (1.0–3.0) and 5.0 (3.0–6.0) for non-ICU and ICU patients, respectively (p<0.001).

**Table 2. Patients' characteristics.**

| Variable | N = 109 |
|---|---|
| Age (mean ± SD) | 60.9 (13.6) |
| Males (n,%) | 76 (69.7) |
| BMI (mean ± SD) | 28.0 (5.4) |
| Smoking (n,%) | 24 (22) |
| Diabetes (n,%) | 35 (32.1) |
| Chronic obstructive pulmonary disease (n,%) | 12 (11) |
| Malignancy (n,%) | 10 (9.2) |
| Congestive heart failure (n,%) | 2 (1.8) |
| Cerebrovascular disease (n,%) | 9 (8.3) |
| Chronic kidney disease (n,%) | 13 (11.9) |
| ICU admission (n,%) | 36 (33) |
| ARDS (n,%) | 27 (24.8) |
| Mechanical ventilation (n,%) | 23 (21.1) |
| Composite outcome* | 24 (22) |
| Vasopressors | 23 (21.1) |
| Hospitalization days (median, interquartile range) | 9 (6–17) |
| ICU days (median, interquartile range) | 10 (4–21.7) |
| Mortality (n,%) | 20 (18.3) |

*Mechanical ventilation > 14 days, prone position or in-hospital mortality

ARDS- Acute Respiratory Distress Syndrome

BMI- Body Mass Index

ICU- Intensive Care Unit

An initial PLIS of 0–2 was found in 81% (n = 60) of patients who had a relatively benign course, did not require ICU care, and remained in a non-ICU ward throughout their hospitalization. In contrast, A first PLIS of 3 or more was found in 68.6% (n = 21) of those patients who ultimately clinically deteriorated during their hospital stay requiring ICU hospitalization (Fig 4).

Multivariate GEE models for the study outcomes are presented in Table 4. Model 1 and 2 show that every increase in one PLIS point was associated with a higher risk for ICU admission

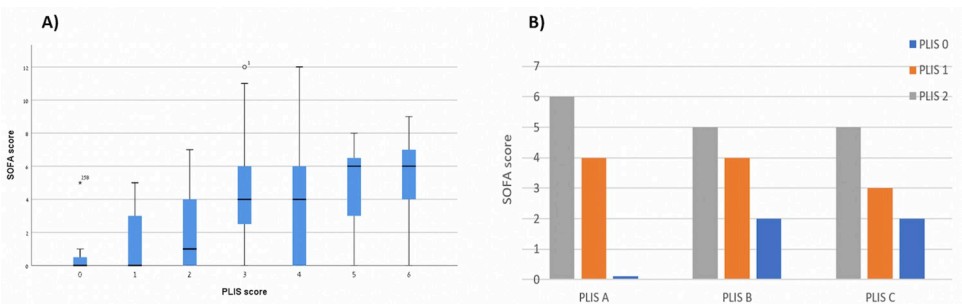

**Fig 2. a:** Presents the association between the SOFA score and each of the PLIS components: PLIS A (P<0.001), PLIS B (p = 0.001), and PLIS C (p = 0.001). **b:** Displays a boxplot diagram of median total PLIS and SOFA scores (interclass correlation 0.63, p<0.001).

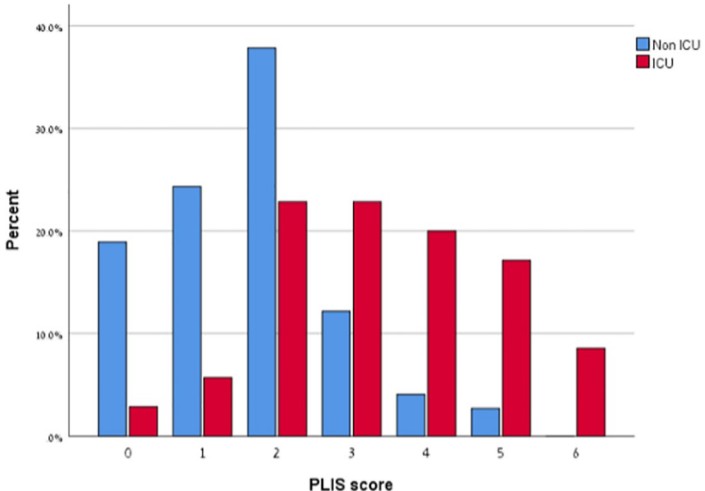

**Fig 3. Shows that both higher initial and worst PLIS during hospitalization are associated with intensive care unit (ICU) admission (A) and in-hospital mortality (B), respectively (P<0.001 for all pairs).**

(O.R 2.09, 95% C.I 1.59–2.75) and in-hospital mortality (O.R 1.54, 95% C.I 1.10–2.16), respectively.

The increase in PLIS point was also associated with increased risk for the composite outcome, composed of prolonged mechanical ventilation over two weeks, prone positioning, or in-hospital mortality (O.R 1.72, 95% C.I 1.24–2.39) (Model 3).

## Discussion

Here we present a new simple clinical-sonographic LUS score that is quick and easy to perform. The PLIS incorporates the level of respiratory support, the extent of the pulmonary interstitial syndrome, and the severity and progression of the consolidation burden in a single and intuitive score. Our main findings illustrate that the PLIS is associated with the patient's clinical status and disease severity. Furthermore, our data suggest that higher PLIS predicts ICU admission and in-hospital death. This method has been efficiently implemented in real-life clinical settings to facilitate daily assessment and follow-up of COVID-19 patients and may predict further patient outcomes and appropriate triage.

COVID-19 patients manifest relative comfort in the face of severe hypoxia, thus masking the severity of their lung injury. However, these patients may rapidly deteriorate without warning. The PLIS offers a straightforward bedside tool that can be done daily as part of the regular follow-up and may help recognize imminent deterioration and pending intubation. The PLIS is a more "friendly" scoring system than those previously referenced, which involves scanning only three fields per lung and a maximal 6 points for ultrasound findings. The PLIS is simple enough to perform in any setting by a lone operator and takes about three minutes to complete by a resident. Six residents and three senior physicians conducted the PLIS after short training with good inter-observer correlation.

The PLIS combines information from patients' LUS findings (B and C scores) with the extent of respiratory support (A score). Similar to the Berlin criteria for Adult Respiratory Distress Syndrome (ARDS), by using the level of hypoxemia for the grading of severity, we believe that combining the extent of respiratory support (PLIS A) with the ultrasonographic score provides another level of information in communicating the patient's health while improving the

**Table 3. Clinical characteristics during PLIS stratified by intensive care unit versus nonintensive care unit.**

| Variable | Non-ICU (n = 103) | ICU (n = 190) | P value |
|---|---|---|---|
| Temperature (mean ± SD) (Celsius) | 36.8 (0.5) | 37.1 (0.7) | <0.001 |
| Oxygen saturation % (mean ± SD) | 94.6 (4.5) | 91.4 (4.8) | <0.001 |
| Mean arterial pressure (mean ± SD) (mmHg) | 89.2 (12.9) | 82.3 (34.7) | 0.02 |
| PaO2/FiO2 (median, interquartile range) * | | 137.1 (95–212.5) | ___ |
| White blood count (median, interquartile range) (K/µL) | 6.4 (5–9.5) | 12 (8.2–17.7) | <0.001 |
| Platelets (median, interquartile range) (K/µL) | 207 (162–281.5) | 320 (230–399) | <0.001 |
| Creatinine (median, interquartile range) (mg/dl) | 0.88 (0.74–1.05) | 0.62 (0.49–1.14) | <0.001 |
| Bilirubin (median, interquartile range) (mg/dl) | 0.53 (0.33–0.67) | 0.40 (0.28–0.61) | 0.03 |
| SOFA score (median, interquartile range) | 0 (0–1) | 5 (4–7) | <0.001 |
| AKI stage (median, interquartile range) | 0 (0–0) | 0 (0–1) | <0.001 |
| PLIS A (n, %) | | | |
| 0 | 72 (69.9) | 27 (14.2) | <0.001 |
| 1 | 31 (30.1) | 63 (66.2) | |
| 2 | 0 (0) | 100 (34.1) | |
| PLIS B (n, %) | | | |
| 0 | 31 (30.1) | 21 (11.1) | <0.001 |
| 1 | 36 (35.0) | 53 (27.9) | |
| 2 | 36 (35.0) | 116 (61.1) | |
| PLIS C (n, %) | | | |
| 0 | 56 (54.4) | 44 (23.2) | <0.001 |
| 1 | 34 (33) | 41 (21.6) | |
| 2 | 13 (12.6) | 105 (55.3) | |
| PLIS score (median, interquartile range) | 2 (1–3) | 5 (3–6) | <0.001 |

AKI- Acute Kidney Injury

ICU- Intensive Care Unit

FiO2- Fractional inspired oxygen

PaO2- Partial pressure of oxygen in arterial blood

PLIS- Point of Care Lung Ultrasound Injury Score

LUS's ability to predict outcomes and prognosis. This is one of the first LUS scores associated with the SOFA score, predicting ICU hospitalization and in-hospital mortality. Multivariate data analysis suggests that for every increase in a single point in the PLIS, ICU admission, in-hospital mortality, and the composite outcome (prolonged mechanical ventilation over two weeks, prone positioning, or in-hospital mortality) increases with odds ratio 2.09, 1.54, and 1.72 respectively.

Furthermore, though this study did not evaluate the use of the PLIS in the outpatient and Emergency Department settings, it is easily conceivable how it can be of value in these settings while guiding triage, referrals, and admitting decisions. In our experience, most non-ICU patients who required hospitalization had a PLIS of 2 or less (Fig 4), suggesting that patients with either a combination of mild interstitial disease with a small consolidation (B1, C1), diffuse severe interstitial disease (B2), or large consolidation (C2), represent disease presentations that warrant careful observation beyond focusing solely on oxygen requirements. Moreover, most non-ICU patients with a relatively benign course that did not necessitate ICU transfer had a PLIS of 2 or less (81%). Conversely, a significant majority of those who did suffer a deleterious course requiring ICU admission had a PLIS of 3 or more (68.6%) (Fig 4). This suggests that the PLIS can serve as a prognostic marker and facilitate triage decisions.

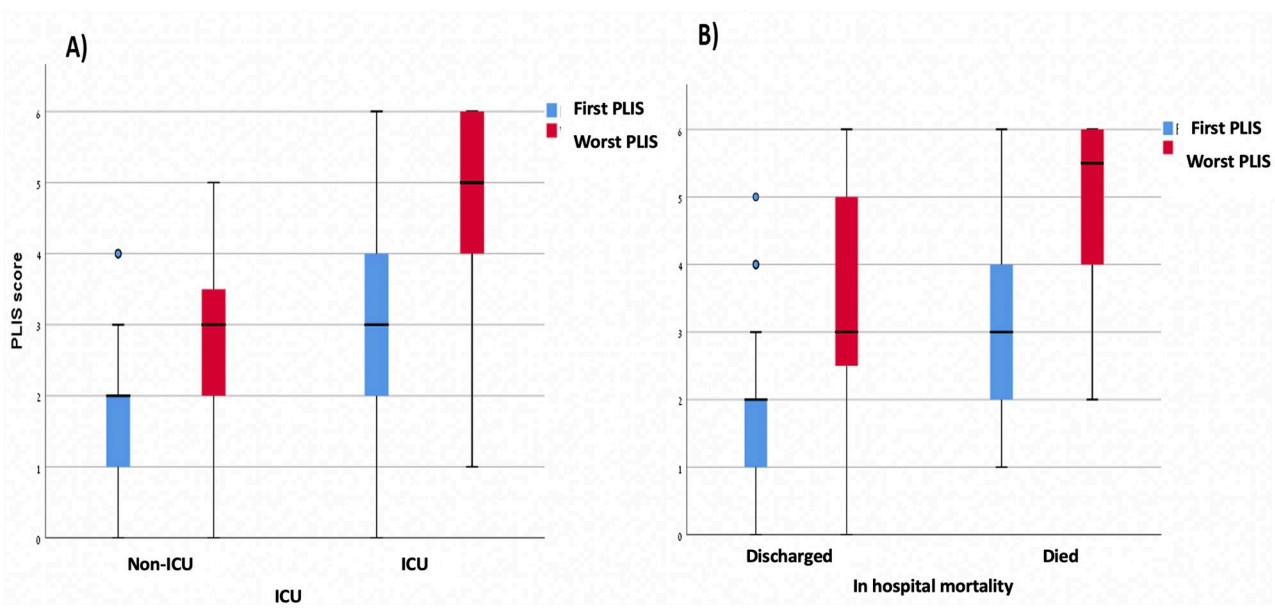

**Fig 4. Presents the results of the initial PLIS stratified by the decision of intensive care unit (ICU) vs. non-ICU admissions (p<0.001).**

The PLIS offers a simple and intuitive three-letter score that illustrates almost graphically an easily conveyable conceptual imaging of the extent of illness within the lungs. While our study did not directly assess the ability of the PLIS to obviate the need for chest X-ray in the management of these patients, well-established literature suggests that in COVID-19 patients, PLIS may diminish the need for frequent chest imaging. Beyond its simplicity, the PLIS addresses an additional limitation of previous LUS scores by providing spatial information as opposed to summing all LUS data into a single discreet number.

The limitations of this report are mainly due to the sample size and its observational nature. Though almost 300 PLIS exams were included, only 109 patients were enrolled; of those, only

**Table 4. Multivariate Generalized Estimating Equation (GEE) regression for study outcomes.**

| | Variable | P value | O.R | 95% C.I |
|---|---|---|---|---|
| Model 1—ICU admission | Age | 0.15 | 1.03 | 0.98–1.09 |
| | Diabetes | 0.03 | 3.49 | 1.12–10.86 |
| | Saturation | 0.01 | 0.88 | 0.8–0.97 |
| | PLIS score | <0.001 | 2.09 | 1.59–2.75 |
| Model 2—in-hospital mortality | Age | 0.01 | 1.09 | 1.02–1.17 |
| | Malignancy | 0.10 | 6.18 | 0.68–55.77 |
| | PLIS score | 0.01 | 1.54 | 1.1–2.16 |
| Model 3—composite outcome* | Age | 0.36 | 1.03 | 0.96–1.11 |
| | Saturation | 0.01 | 0.88 | 0.81–0.96 |
| | AKI | <0.001 | 3.58 | 1.76–7.3 |
| | PLIS score | 0.001 | 1.72 | 1.24–2.39 |

*Mechanical ventilation > 14 days, prone position or in-hospital mortality

AKI- Acute Kidney Injury

ICU- Intensive Care Unit

PLIS- Point of Care Lung Ultrasound Injury Score

36 were in the ICU, all from one tertiary care medical center. Not all patients with COVID-19 admitted to medicine floors (i.e., outside the ICU) received a PLIS, depending on the availability of the trained physicians on call, which could result in unintentional sampling bias. A larger, multicenter study is needed to better validate generalizability and PLIS prediction power for outcomes. We did not evaluate the association between the PLIS phenotype and the potential for lung recruitment, a topic ripe for future study. Could a specific disease pattern, such as a higher B score with a low C score, suggest positive end-expiratory pressure, prone positioning, or inhaled nitric oxide responsiveness? Such data is essential if this score is to be a dynamic, daily management tool that can aid in clinical decisions. Lastly, the studied group may not reflect the prevalence of underlying diseases in other countries or ethnicities. For example, CKD or CHF may be more prevalent in certain populations which potentially can affect the lung sonographic profile.

## Conclusions

We introduce a novel lung ultrasound score for COVID-19 patients that is simple, illustrative, and associated with SOFA score, ICU admission, and in-hospital mortality. Further studies are needed to demonstrate if PLIS can become an integral part of the daily assessment and management of COVID 19 patients and ARDS patients in general.

## Author Contributions

**Conceptualization:** Lior Fuchs, Ori Galante, Yaniv Almog, Roy R. Dayan.

**Data curation:** Lior Fuchs, Yuval Ullman, Ran Ben David Ohayon, Evgeny Golbets, Khaled El Haj, Itai Weissberg, Leonid Barski, Eli Rosenberg, Eyal Gohar, Muhammad Abo Abed, Iftach Sagy.

**Formal analysis:** Victor Novack, Iftach Sagy.

**Investigation:** Lior Fuchs, Ori Galante, Roy R. Dayan, Alexander Smoliakov, Yuval Ullman, David Shamia, Ran Ben David Ohayon, Eyal Gohar, Muhammad Abo Abed.

**Methodology:** Lior Fuchs, Ori Galante, Yuval Ullman, Jonathan Taylor, Iftach Sagy.

**Project administration:** Lior Fuchs.

**Supervision:** Lior Fuchs.

**Validation:** Jonathan Taylor, Victor Novack, Leonid Barski.

**Visualization:** Khaled El Haj.

**Writing – original draft:** Lior Fuchs, Roy R. Dayan, David Shamia, Jonathan Taylor, Iftach Sagy.

**Writing – review & editing:** Lior Fuchs, Yaniv Almog, Roy R. Dayan.

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
