## [Decision Letter · Decision Letter 0]

17 Mar 2022

PONE-D-21-38198

Point of Care Lung Ultrasound Injury Score - a Simple and Reliable Assessment Tool in COVID-19 Patients (PLIS I): A Retrospective Study

PLOS ONE

Dear Dr. Fuchs,

Thank you for submitting your manuscript to PLOS ONE. After careful consideration, we feel that it has merit but does not fully meet PLOS ONE’s publication criteria as it currently stands. Therefore, we invite you to submit a revised version of the manuscript that addresses the points raised during the review process.

We look forward to receiving your revised manuscript.

Kind regards,

Jignesh K. Patel

Academic Editor

PLOS ONE

Journal Requirements:

2. Thank you for stating the following financial disclosure: "this is not a funded research"

3. Thank you for stating the following in the Competing Interests section: "Lior Fuchs is a consultant for GE Healthcare"

5. Please amend the manuscript submission data (via Edit Submission) to include author Khaled el haj, MD.

7. Please upload a new copy of Figures 2 and 3 as the detail is not clear. Please follow the link for more information: https://blogs.plos.org/plos/2019/06/looking-good-tips-for-creating-your-plos-figures-graphics/" https://blogs.plos.org/plos/2019/06/looking-good-tips-for-creating-your-plos-figures-graphics/

Reviewers' comments:

Reviewer's Responses to Questions

**Comments to the Author**

1. Is the manuscript technically sound, and do the data support the conclusions?

Reviewer #1: Yes

Reviewer #2: Yes

2. Has the statistical analysis been performed appropriately and rigorously? 

Reviewer #1: Yes

Reviewer #2: Yes

3. Have the authors made all data underlying the findings in their manuscript fully available?

Reviewer #1: Yes

Reviewer #2: Yes

4. Is the manuscript presented in an intelligible fashion and written in standard English?

Reviewer #1: Yes

Reviewer #2: Yes

5. Review Comments to the Author

Reviewer #1: Thanks for this interesting paper on the use of ultrasound in COVID patients.

My comments are :

Background : could you provide some references to “There is a direct association between the severity of COVID-19 pneumonia and computerized tomography (CT) findings”

Could you add something about the predominant COVID variant in your hospital during the study?

Could you please address this comment: This is a retrospective study, although data were prospectively collected.

Was the study designed prospectively?? Was it registered? Could you discuss how the study was designed?

An important question is the timing of the data collection. The COVID time course is useful in understanding deterioration, and I am not sure from the manuscript whether the measures on patients reflect – paired measurements (ED or ward then ICU, or in some patients were single measurements, or in others, multiple measurements on the same patient). It just states “on admission” – not sure whether that is to hospital or the ward or to ICU.

P 5 “were assessed by an intensivist to require mechanical ventilation in the next 24 hours potentially. “

Were assessed by an intensivist to potentially require …

Figure 1: why are the stars in a different position to the probe?

For the “putting it together” could you create a figure that shows perhaps the different components with the ultrasound images ?

Interobserver variability – how many scans were repeated by different operators?

Table 4 Did you run the model just with PLIS A? Just wanting to understand the added benefit of the U/S scan in predictions.

Reviewer #2: This is a interesting paper which validates some of our regular utility of sonography in the ICU during the pandemic. We have already appreciated that CT scan findings with higher "scores" i.e. worse diseases, likely have a poorer outcomes. This essentially is also shown in this paper where the authors evaluate the lung parenchyma.

Using PLIS in their group is sound however this group may not reflect the prevalence of underlying diseases in other countries and so this should be taken with a grain of salt. For example, CKD or CHF may be more prevalent in certain populations which would affect the lung sonographic profile.

6. PLOS authors have the option to publish the peer review history of their article (what does this mean?). If published, this will include your full peer review and any attached files.

Reviewer #1: No

Reviewer #2: **Yes: **Kinner Patel

---

## [Author Response · Author response to Decision Letter 0]

2 Apr 2022

Answer letter to: 

PONE-D-21-38198

Point of Care Lung Ultrasound Injury Score - a Simple and Reliable Assessment Tool in COVID-19 Patients (PLIS I): A Retrospective Study

PLOS ONE

Dear Editor, 

Thank to you and to the reviewers, we believe that the manuscript now is better, more clear to the readers, and improved:

Answer: Done 

2. answer: “The authors received no specific funding for this work.”

Answer: Done 

3. Thank you for stating the following in the Competing Interests section: "Lior Fuchs is a consultant for GE Healthcare."

Answer: I – Lior Fuchs, declare that: "This does not alter our adherence to PLOS ONE policies on sharing data and materials.”

It is updated in the cover letter.

Answer: unfortunately, there are ethical and legal restrictions to sharing our data publicly:

Based on Clalit Health Services regulations, the study data set maintain only at Clalit Health Services servers. Researchers are not allowed to upload or share the data set with other sources.

5. Please amend the manuscript submission data (via Edit Submission) to include author Khaled el haj, MD:

Answer: Done

Answer: Done

7. Please upload a new copy of Figures 2 and 3 as the detail is not clear. Please follow the link for more information: https://blogs.plos.org/plos/2019/06/looking-good-tips-for-creating-your-plos-figures-graphics/" https://blogs.plos.org/plos/2019/06/looking-good-tips-for-creating-your-plos-figures-graphics/

Answer: Done

Answer: Sorry for the inconsistency with the references. There was a technical problem with the order of the references. I have reviewed the reference list and corrected it. No references were retracted. All original references are labeled by the PLOS one font instructions. 

Reviewers' comments:

Reviewer's Responses to Questions

Comments to the Author

1. Is the manuscript technically sound, and do the data support the conclusions?

Reviewer #1: Yes

Reviewer #2: Yes

2. Has the statistical analysis been performed appropriately and rigorously? 

Reviewer #1: Yes

Reviewer #2: Yes

3. Have the authors made all data underlying the findings in their manuscript fully available?

Reviewer #1: Yes

Reviewer #2: Yes

4. Is the manuscript presented in an intelligible fashion and written in standard English?

Reviewer #1: Yes

Reviewer #2: Yes

5. Review Comments to the Author

Reviewer #1: Thanks for this interesting paper on the use of ultrasound in COVID patients.

My comments are :

Background : could you provide some references to “There is a direct association between the severity of COVID-19 pneumonia and computerized tomography (CT) findings.”

Answer: there were errors with the references labels – now we provide the right references at the right place, as well as the one in your comment. 

Could you add something about the predominant COVID variant in your hospital during the study?

Answer: The study period included the Alpha, and Beta SARS-CoV-2 variants.

We added this sentence to the first paragraph of the Methods. 

Could you please address this comment: This is a retrospective study, although data were prospectively collected.

Was the study designed prospectively?? Was it registered? Could you discuss how the study was designed?

Answer: we agree that the term “prospectively collected “ is confusing. This is a retrospective study that analyses retrospectively a new clinical lung US score that was implemented during the COVID 19 outbreak. We removed the term prospectively collected and added it to the methods section: 

“This is a retrospective study; data were collected during patients assessments and rounds. The PLIS protocol was taught and implemented in the COVID 19 wards and ICUs and was part of the daily patients assessment.”

Later in Methods, we further explain the way we collected the data: “Data collected included the PLIS recorded on admission, when conducted during ward/ICU rounds, and when otherwise clinically indicated.”

This was a routine lung assessment for all. All about it is a retrospective analysis. 

An important question is the timing of the data collection. The COVID time course is useful in understanding deterioration, and I am not sure from the manuscript whether the measures on patients reflect – paired measurements (ED or ward then ICU, or in some patients were single measurements, or in others, multiple measurements on the same patient). It just states “on admission” – not sure whether that is to hospital or the ward or to ICU.

Answer: Thank you for the important comment. For clarification: this is not a prospective study, and PLIS exams were not protocolized. Some of the patients received only ward PLIS assessments as they were never admitted to the ICU. Some received only in the ICU and not in the ward. Some were received in both locations. The studies were not paired. 

We added the following sentence to the second paragraph in the Methods: 

“As this is not a prospective study, the PLIS exams were not protocolized. Some patients could receive a PLIS study on regular ward admission, during the morning rounds, and also later, when admitted to the ICU. Some received ultrasound lung scan only during the ICU admission or only in the regular ward.”

P 5 “were assessed by an intensivist to require mechanical ventilation in the next 24 hours potentially. “

Were assessed by an intensivist to potentially require …

Answer: Thanks, we have added “potentially”. 

Figure 1: why are the stars in a different position to the probe?

Answer:The stars represent the area of scan in each zone and not the exact place of the probe. This comment was added to the Figure 1 legend. 

For the “putting it together,” could you create a figure that shows perhaps the different components with the ultrasound images ?

Answer: Thank you for an excellent idea. Figures 1b and 1c with b lines and consolidation were added and referred to in the putting it together paragraph. 

Interobserver variability – how many scans were repeated by different operators?

Answer: Inter-observer reliability was assessed by comparing independent ratings of the PLIS B and C scores among 16 randomly selected patients in a blinded manner by two physicians. Please see the Statistical Analysis section for more details.

Table 4 Did you run the model just with PLIS A? Just wanting to understand the added benefit of the U/S scan in predictions.

Answer: PLIS A is connected to the prognosis, as we know from previous data regarding the outcomes of SARS- 2 ventilated patients. This is the reason we present Figure 2, which shows the association between the SOFA, PLIS B and PLIS C. We also analyzed the association between these lung ultrasound scores (PLIS B and C, independently from PLIS A) and other outcomes, as the risk for mechanical ventilation. We found OR of over 4 and over 10 for PLIS B2 and PLIS C2, respectively, compared to PLIS B0 and C0. So, we do know that the ultrasound score is a predictor and associated with bad outcomes, regardless of the PLIS A score. We chose to publish these specific associations in PLIS II report. 

Reviewer #2: This is an interesting paper which validates some of our regular utility of sonography in the ICU during the pandemic. We have already appreciated that CT scan findings with higher "scores" i.e. worse diseases, likely have a poorer outcomes. This essentially is also shown in this paper where the authors evaluate the lung parenchyma.

Using PLIS in their group is sound however this group may not reflect the prevalence of underlying diseases in other countries and so this should be taken with a grain of salt. For example, CKD or CHF may be more prevalent in certain populations which would affect the lung sonographic profile.

Answer: I strongly concur. We have added this sentence to the Limitations paragraph. 

6. PLOS authors have the option to publish the peer review history of their article (what does this mean?). If published, this will include your full peer review and any attached files.

Do you want your identity to be public for this peer review? For information about this choice, including consent withdrawal, please see our Privacy Policy.

Reviewer #1: No

Reviewer #2: Yes: Kinner Patel

---

## [Editor Report · Decision Letter 1]

11 Apr 2022

Point of Care Lung Ultrasound Injury Score - a Simple and Reliable Assessment Tool in COVID-19 Patients (PLIS I): A Retrospective Study

PONE-D-21-38198R1

Dear Dr. Fuchs,

We’re pleased to inform you that your manuscript has been judged scientifically suitable for publication and will be formally accepted for publication once it meets all outstanding technical requirements.

Kind regards,

Jignesh K. Patel

Academic Editor

PLOS ONE

Additional Editor Comments (optional):

author's have appropriately answer and modify the manuscript accordingly.
---

## [Editor Report · Acceptance letter]

14 Apr 2022

PONE-D-21-38198R1 

Point of Care Lung Ultrasound Injury Score - a Simple and Reliable Assessment Tool in COVID-19 Patients (PLIS I): A Retrospective Study 

Dear Dr. Fuchs:

I'm pleased to inform you that your manuscript has been deemed suitable for publication in PLOS ONE. Congratulations! Your manuscript is now with our production department. 

Kind regards, 

on behalf of

Dr. Jignesh K. Patel 

Academic Editor

PLOS ONE